# *Pf*Ago-Based Zika Virus Detection

**DOI:** 10.3390/v16040539

**Published:** 2024-03-30

**Authors:** Yuhao Chen, Xianyi Zhang, Xuan Yang, Lifang Su, Weiran Chen, Jixiang Zhao, Yunhong Hu, Yuan Wang, Ying Wu, Yanming Dong

**Affiliations:** 1State Key Laboratory of Biocatalysis and Enzyme Engineering, Hubei Key Laboratory of Industrial Biotechnology, Hubei Key Laboratory of Industrial Biotechnology and Hubei Collaborative Innovation Center for Green Transformation of Bio-Resources, School of Life Sciences, Hubei University, Wuhan 430062, China; 2School of Basic Medicine, Hubei University of Arts and Sciences, Xiangyang 441053, China; 3State Key Laboratory of Virology and Hubei Province Key Laboratory of Allergy and Immunology, Institute of Medical Virology, TaiKang Medical School (School of Basic Medical Sciences), Wuhan University, Wuhan 430072, China; 4Hubei Jiangxia Laboratory, Wuhan 430200, China

**Keywords:** nucleic acid detection, Zika virus, *Pyrococcus furiosus* Argonaute protein, gDNA

## Abstract

As a mosquito-borne flavivirus, Zika virus (ZIKV) has been identified as a global health threat. The virus has been linked to severe congenital disabilities, including microcephaly and other congenital malformations, resulting in fatal intrauterine death. Therefore, developing sensitive and specific methods for the early detection and accurate diagnosis of the ZIKV is essential for controlling its spread and mitigating its impact on public health. Herein, we set up a novel nucleic acid detection system based on *Pyrococcus furiosus* Argonaute (*Pf*Ago)-mediated nucleic acid detection, targeting the non-structural protein 5 (NS5) region of the ZIKV genome (abbreviated ZIKV-PAND). Without preamplification with the polymerase chain reaction (PCR), the minimum detection concentration (MDC) of ZIKV-PAND was about 10 nM. When introducing an amplification step, the MDC can be dramatically decreased to the aM level (8.3 aM), which is comparable to qRT-PCR assay (1.6 aM). In addition, the diagnostic findings from the analysis of simulated clinical samples or Zika virus samples using ZIKV-PAND show a complete agreement of 100% with qRT-PCR assays. This correlation can aid in the implementation of molecular testing for clinical diagnoses and the investigation of ZIKV infection on an epidemiological scale.

## 1. Introduction

Zika virus (ZIKV), which belongs to the *Flavivirus* genus of the *Flaviviridae* family, is a small, enveloped, positive-stranded RNA virus and consists of a single-stranded 11 kb RNA genome. Its genome encodes a polyprotein categorized into three structural proteins (capsid (C), pre-membrane (prM), and envelope (E)) and seven non-structural proteins (NS1, NS2A, NS2B, NS3, NS4A, NSB, and NS5) [1]. As a mosquito-borne flavivirus, the virus transmits primarily through *Aedes aegypti* mosquitoes and can also transmit through sexual intercourse, blood transfusion, and from mother to infant [2,3]. ZIKV infection tends to be asymptomatic or mildly symptomatic and even can cause more severe clinical diseases, such as Guillain–Barré syndrome in adults and children. Moreover, viral infection during pregnancy can cause congenital Zika syndrome (CZS), including microcephaly and other congenital malformations that result in intrauterine fatal death [4,5,6]. In 2016, the World Health Organization categorized ZIKV as an international concern for public health emergency (PHEIC). Controlling the spread of ZIKV calls for the early detection and accurate diagnosis of this etiological agent. Within the various viral proteins, the viral NS5 protein performs essential roles in viral genome replication, capping, and interferon suppression, providing it a unique target for discovering potential drug-like candidates with high specificity and low toxicity [7,8]. Notably, the NS5 protein is the most conserved ZIKV protein, exhibiting a 94% sequence similarity across the two primary ZIKV lineages—Asian and African [7]. Given its elevated expression levels compared to other viral proteins, NS5 emerges as a promising candidate for enhancing sensitivity in molecular diagnostics [9].

For the detection of ZIKV, viral isolation from clinical samples or cell cultures has been considered the “gold standard”, but it has been replaced by molecular tests in most clinical virology labs due to limitations such as being limited to detecting cultivable viruses, having a low yield, and lower sensitivity [10]. Currently, more serological assays, such as the IgM antibody capture enzyme-linked immunosorbent assay (MAC-ELISA), plaque reduction neutralization test (PRNT), immunofluorescence assay (IFA), reporter virus neutralization test (RVNT), and multiplex microsphere immunoassays (MIA) based on the detection of antigens and antibodies, have been developed for ZIKV detection [11,12,13,14,15]. However, ZIKV antibodies used in the afore-mentioned assays could have low sensitivity and specificity, and also exhibit cross-reactivity against other homologous flaviviruses, such as the dengue virus (DENV) [10,16]. Hence, additional advancements in serological measurements are necessary to address the performance limitations. Because of the exceptional sensitivity and specificity, nucleic acid amplification tests (NAATs), like RT-PCR, qRT-PCR, Pan-flavivirus RT-PCR, nested RT-PCR, and droplet digital PCR (ddPCR), have significantly contributed to the verification of ZIKV detection [16]. Due to their rapid processing speed, as well as their heightened sensitivity and specificity, molecular techniques have become the predominant diagnostic method for most viral CNS infections [10,17]. As an accurate and rapid biosensor, the CRISPR/Cas system has expanded to a new use of diagnostics for infectious diseases, including the detection of various viral nucleic acids [18,19,20,21]. For ZIKV detection, the Cas13-based SHERLOCK (specific high-sensitivity enzymatic reporter unlocking) platform can probe and distinguish ZIKV and four dengue virus (DENV) serotypes in infected human-patient bodily fluid samples at concentrations as low as one copy per microliter [22]. To avoid the high cost of guide RNA (gRNA) synthesis and alleviate the dependence of the protospacer-adjacent motif (PAM) in the CRISPR/Cas system, we recently developed a cost-effective *Pf*Ago-mediated nucleic acid detection (PAND) technology, which has been applied in different viral nucleic acid detections, including human papillomavirus 16 (HPV-16), severe acute respiratory syndrome coronavirus 2 (SARS-CoV2), human parvovirus B19, and African swine fever virus (ASFV) [23,24,25]. During the procedure of PAND, three 5′-phosphorylated single-stranded guide DNAs (5’P-gDNAs) directed *Pf*Ago to cleave the ZIKV target DNA and thus spawn a 16nt 5′-phosphorylated single-stranded DNA (ssDNA), which, in turn, served as the new second-round gDNA bonded to the apo form of *Pf*Ago and incised its complementary molecular beacons, leading to the split of the quenchers from each other and subsequent fluorescence detection.

This study established a unique ZIKV viral nucleic acid detection system based on the PAND method. In addition, the diagnostic results for the sample clinical simulation with ZIKV-PAND reveal 100% consistency with qRT-PCR assays, which can assist in molecular testing for the clinical diagnosis and epidemiological investigations of ZIKV infection.

## 2. Materials and Methods

### 2.1. Cells and Virus

The human embryonic kidney cells (HEK293T), Vero cells, were cultured in Dulbecco’s Eagle medium (DMEM) supplemented with 10% (*v*/*v*) FBS and 1% (*v*/*v*) Penicillin Streptomycin incubating in an atmosphere at 37 °C containing 5% CO_2_. The Zika virus (GenBank accession number: MK829154) was propagated in Vero cells, as previously described [26].

### 2.2. PfAgo Endonuclease Cleavage Activity Assay

The His-*Pf*Ago protein was expressed in *Escherichia coli* BL21 and purified using Ni-NTA affinity as described previously [25]. And the endonuclease cleavage assay of *Pf*Ago was performed as follows: briefly, 2 pmol of indicated 5′g-DNA, 5′ phosphorylated by T4 polynucleotide kinases (T4 PNK), and 0.5 pmol of ssDNA (target ssDNA/ZIKV-ssDNA/MB-ZIKV/f-MB-ZIKV) or NS5 DNA PCR product targets were incubated with or without the His-*Pf*Ago fusion protein (45 pmol) in a total volume of 20 μL reaction buffer (20 mM HEPES pH 7.5, 250 mM NaCl, and 0.5 mM MnCl_2_) at 95 °C for 20 min. The cleaved product was then analyzed by 20% TBE-PAGE electrophoresis and followed by SYBR gold nucleic acid dye staining, as previously described [27]. When using molecular beacon f-MB-ZIKV, the fluorescence intensity was measured using blue-light transilluminators or microplate readers.

### 2.3. The Sensitivity of ZIKV-PAND Detection

For the sensitive detection of ZIKV-PAND without PCR, a serial amount of the ZIKV NS5 target PCR product (achieved with NS-F/R through PCR amplification) was adjusted to a final concentration of 50 nM, 30 nM, 25 nM, 20 nM, 15 nM, 10 nM, 5 nM, 3 nM, 1 nM, or 0.5 nM and followed by ZIKV-PAND detection, as mentioned above. For the ZIKV-PAND sensitivity assay with PCR, an initial concentration of 6.73 × 10^8^ aM (2.03 × 10^7^ copies/μL) ZIKV plasmid (pUC-ZIKV) was determined and followed by a serial 10-fold dilution to a final concentration of 6.73 × 10^8^ aM to 6.73 × 10^−1^ aM (2.03 × 10^−2^ copies/μL). Furthermore, the final concentration from 50, 5, 0.5, 0.4, 0.2 to 0.05 copies/μL of pUC-ZIKV was also determined. Then, 1 μL of the indicated ZIKV plasmid was used as a template, and the ZIKV NS5 target PCR product was created with NS-F/R through PCR amplification in a total 10 μL reaction volume. Finally, ZIKV-PAND detection was measured as mentioned above. The details of the indicated plasmids, primers, PCR program, gDNA, and ssDNA targets utilized in the ZIKV-PAND assays are included in Appendix A.

### 2.4. The Specificity of ZIKV-PAND Detection

For the specificity of ZIKV-PAND detection, the conserved NS5 target product of DENV1, as shown in Appendix A, was firstly synthesized, and cloned into pUC-18. Then the PCR product from pUC-DENV1 and pUC-ZIKV was amplified and used as the target for the ZIKV-PAND assay, as described above.

### 2.5. Analysis of Sample Simulation or Wildtype ZIKV with ZIKV-PAND

For the analysis of the sample simulation with ZIKV-PAND, CMV-promoter driven constructs (ZIKV-EGFP/DENV-EGFP) were firstly produced and transfected into HEK293T cells. After the RNA extraction of the total cells using TRIzol, cDNA synthesis was performed with the indicated ZIKV NS5 primer, which was employed for the subsequent ZIKV-PAND and qRT-PCR assays. For the detection of wildtype ZIKV, Vero cells were infected with ZIKV, and the supernatant containing the viral particles was collected at 2 h and 30 h. And the Zika virus genome RNA was extracted with the TIANamp Virus RNA kit (TIANGEN). ZIKV cDNA was reverse-transcribed and employed for the qRT-PCR assay; ZIKV-PAND was coupled with PCR.

## 3. Results

To establish the ZIKV-PAND system, as shown in Appendix A, the recombinant His-PfAgo protein possessing the endonuclease activity mediated by gDNA was first purified, as described previously [25]. As shown in Figure 1, three gDNAs (gr, gt, and gf) targeting the ZIKV *ns*5 conserved region and gn (fMB-ZIKV) targeting the molecular beacon were designed. Following the RT-PCR procedure, the endonuclease *Pf*Ago-mediated digestion assay was performed by incubating at 95 °C for 20–30 min after adding 5’P-gDNAs and a molecular beacon, as described previously [25]. To create a ZIKV-PAND reaction, three specific 5’P-gDNAs can guide *Pf*Ago to cleave cDNA, which are reverse-transcribed from ZIKV genomic RNA and thus produce 16nt 5′-phosphorylated ssDNA. This newly created product in turn can serve as a second-round gDNA to form a complex with the apo form of *Pf*Ago and then cleave its complementary DNA beacons, leading to the separation of quenchers from each other and subsequent fluorescence detection. In the absence of target DNA for the various gDNAs (gr, gt, and gf) to recognize and cleave, no fluorescence is observable as a control measure.

To direct *Pf*Ago to target ZIKV DNA and the molecular beacon (fMB-ZIKV), MB selection was first performed with *Pf*Ago-mediated cleavage. We identified an applicable non-fluorescent MB that could be cleaved to produce 16 nt ssDNA by *Pf*Ago-mediated cleavage with gDNA(gn) (Appendix A). As described above, we subsequently performed the endonuclease activity assay using *Pf*Ago and three gDNAs (gr, gt, or gf) that targeted the ZIKV *ns*5 region. As expected, following the ZIKV-PAND reaction, 249 bp of target ZIKV DNA (ZIKV NS5-P) was cleaved to produce a 16nt 5′P-gn DNA, which initiated the second-round cleavage of the fluorescent molecular beacon (f-MB-ZIKV) (Figure 2a, Lane 2, and Lane 4). Moreover, the fluorescence intensity detection result shows that the cleavage activity of *Pf*Ago is 34-times higher than that of the control (Figure 2b, left panel). In parallel, the significant fluorescence was observed in the ZIKV-PAND reaction, but not in the control system without *Pf*Ago using blue-light transilluminators (Figure 2b, right panel). These results indicate that ZIKV-PAND was established and can potentially be used for ZIKV detection. 

To evaluate the MDC and sensitivity of ZIKV-PAND, preamplification with or without a PCR was compared in this system. As seen on the left panel in Figure 2c, the MDC of ZIKV-PAND without NS5 preamplification is 10 nM (10 fmol/µL), which is consistent with our previous research [28]. As shown in Figure 2d and Appendix A, when inducing preamplification with PCR, ZIKV-PAND can detect an efficient concentration of the target down to ~8.3 aM (5.0 copies/µL) that is comparable to the MDC for qPCR methods (1.67 aM/1.0 copies/µL) with SYBR green (Appendix A). ZIKV and DENV both share the same *Aedes* (*Stegomyia*) mosquito vectors and geographic distributions, but these infections cannot be easily distinguished clinically. Therefore, for a ZIKV nucleic acid diagnosis, false-positive signaling is a common problem for susceptible detection methods, such as qRT-PCR and RT-LAMP assays [13]. To investigate the specificity of ZIKV-PAND, we first chose to clone the DENV1 construct (shown in Appendix A) to obtain the PCR product target for subsequent PAND. The results show that no fluorescence can be observed in the ZIKV-PAND test when replacing the ZIKV NS5 target with DENV1 NS5 (Figure 2e, Lane 2). Moreover, no dramatic influence on the ZIKV target detection can be observed when mixing ZIKV and DENV1 NS5 PCR products (Figure 2e, Lane 1). Consistently, a similar result was obtained via the qPCR test (Appendix A). Notably, the Ct value was 30 when replacing the ZIKV NS5 target with DENV1. This means that the above-mentioned ZIKV qPCR primers also show some cross-reactivity with DENV1 DNA. Consequently, although further specificity optimization of the above-mentioned ZIKV qPCR primers should be performed, these results still suggest that ZIKV-PAND detection has similar analytical sensitivities compared to the RT-qPCR assay.

Moreover, we performed a ZIKV nucleic acid test with a sample of clinical simulations using a ZIKV-PAND system coupled with RT-PCR. After the transfection of HEK293T cells with CMV-promoter driven constructs (ZIKV-EGFP/DENV1-EGFP) containing ZIKV and the DENV1 target region, an EGFP expression was observed, which meant that the target ZIKV and DENV1 RNA were transcribed and could be used as simulated samples (Appendix A). And the results of the fluorescence test show that the sample containing the ZIKV target RNA but not DENV1 RNA exhibits cleavage when using ZIKV-PAND coupled with RT-PCR (Figure 2f). Meanwhile, as we expected, the detection efficiency of this method is also in 100% concordance with the RT-PCR measurement (Appendix A). Furthermore, in cases where extra non-specific viral DNA, such as Rubella virus (RV) and Cytomegalovirus (CMV), replaced the ZIKV target DNA within the ZIKV-PAND setup, no cleavage could be identified, implying that our assay panel displayed remarkable specificity (Appendix A). Additionally, we performed a ZIKV nucleic acid test with wildtype ZIKV particles using ZIKV-PAND systems coupled with RT-PCR. The results show that ZIKV RNA has been detected in supernatant taken from 2 h or 30 h ZIKV-infected Vero cells, but not in the mock infection and negative control (Figure 2g). In our attempt to further determine whether the wildtype ZIKV detection with ZIKV-PAND was in accordance with above MDC test results, we first chose to test the viral load of ZIKV (30 h ZIKV-infected cells) using a qRT-PCR assay. And the results show that there is approximately 94.8 copies/µL of the ZIKV RNA genome (Appendix A). Finally, as is shown in Figure 2h, 15.75 aM (9.48 copies/µL) of sensitivity for ZIKV genome detection can be observed using ZIKV-PAND coupled with RT-PCR, which is consistent with the ~8.3 aM (5.0 copies/µL) sensitivity that we tested above. Together, these results demonstrate that ZIKV-PAND combined with RT-PCR has high sensitivity and specificity, which can be beneficial in the clinical diagnosis of ZIKV infection.

## 4. Conclusions

Since the outbreak of 2013/14, ZIKV remains a significant public health threat. No specific antiviral drugs and vaccines are available to treat and prevent the viral infection [5]. Therefore, developing quick and precise diagnostic tools for ZIKV infection remains an active area of research. In comparison with conventional diagnostic methods, such as the qPCR test, programmable biosensors based on CRISPR or pAgos diagnostic systems have been utilized as potential diagnostic tools due to their high specificity and efficiency for nucleic acid targets [29,30]. Here, we developed a novel nucleic acid detection system called ZIKV-PAND, which specifically targets the NS5 region of the ZIKV RNA genome, using the *Pf*Ago protein. Remarkably, without preamplification with PCR, gDNA-mediated ZIKV-PAND had an MDC of 10 nM. And when introducing the RT-PCR step for ZIKV target amplification, a 5.0 copies/µL level of MDC was measured, which is approximately 10-fold higher than that of previous *Pf*Ago- or *Ttr*Ago-based methods [27,28]. Furthermore, the sensitivity achieved by ZIKV-PAND was similar to that of the ZIKV qRT-PCR assay with the same primer set as previously mentioned. However, ZIKV-PAND does not need sophisticated laboratory equipment or experienced operators compared to the qPCR test, which can reduce the cost and complexity of the detection. As a mosquito-borne flavivirus, the symptoms caused by ZIKV are difficult to distinguish from other arboviruses. The serological assays for ZIKV detection are highly challenging due to cross-reactivity. Therefore, the timely and accurate diagnosis of ZIKV infection is crucial for selecting the appropriate treatment strategies. In this report, it has been demonstrated that the gDNA in ZIKV-PAND is unique for detecting ZIKV and does not cross-react with the DENV1 genome. This means that ZIKV-PAND has higher specificity for nucleic acid diagnostics. Currently, several diagnostic platforms, including NASBACC (Cas9), SHERLOCK (Cas13a), DETECTR (Cas12a), or Cas4-DETECTR (Cas14a), for nucleic acid detection with the trans-cleavage of active CRISPR-Cas have been established and exploited for pathogen nucleic acid detection [31]. Among these platforms, SHERLOCK and DETECTR have received FDA Emergency Use Authorization (EUA). Moreover, the Cas12a-mediated DETECTR method has a detection limit of approximately 20 copies per microliter, whereas SHERLOCK relies on Cas13a and can detect pathogens at concentrations as low as approximately 1 copy per microliter [22,31]. Meanwhile, SHERLOCK enables instrument-free ZIKV and DENV detection directly from patient samples in <2 h [22]. When compared to the SHERLOCK and DETECTR methods, our ZIKV-PAND method also demonstrated similar high sensitivity and specificity when detecting viral nucleic acid. Significantly, ZIKV-PAND offers enhanced affordability and simplicity in choosing a responsive guide DNA. This is achieved by removing the need for costly guide RNA (gRNA) and by eliminating the reliance on the protospacer-adjacent motif (PAM). Undoubtedly, this approach presents certain unresolved issues. While PAND tests without amplification usually last 20–30 min, conventional RT-PCR tests typically take 45 to 90 min. Therefore, the ZIKV-PAND combined RT-PCR test is expected to require approximately 65 to 120 min, which is more time-consuming compared to the SHERLOCK and DETECTR methods utilizing isothermal amplifications [31]. As a result, these factors impose restrictions on its widespread implementation. Furthermore, additional research will be carried out to merge ZIKV-PAND with isothermal amplification techniques, thereby decreasing the detection time and removing the necessity for specialized equipment and expertise. Furthermore, the future clinical diagnoses will validate the ability of *Pf*Ago to differentiate single-nucleotide mutants. This will allow for the simultaneous detection of arboviruses such as ZIKV, DENV, and chikungunya in a single-tube reaction. It has been shown that certain host proteins, like TRIM56, are capable of binding to ZIKV RNA, potentially influencing the binding and cleavage preferences of RNA-cleaving proteins, such as *Km*Ago [32,33]. This influence should be taken into consideration when employing these pAgos for targeting and cleaving ZIKV RNA within cells. As the most-conserved ZIKV protein, NS5 protein exhibits a 94% sequence similarity across the two primary ZIKV Asian and African lineages [7]. Moreover, as shown in Appendix A, the nucleotide sequence comparisons of the ZIKV NS5 region targeting gr/gf/gt reveal that gf and gr are fully consistent with both lineages, while there is a single discrepancy between gt and the Asian lineage. This suggests that our ZIKV-PAND is more effective in identifying ZIKV African lineages when utilizing these three different genomic DNA samples. As a PAND assay using a single-genomic DNA has also been developed, optimizing the use of gf for this single-genomic DNA can enhance the detection of the primary ZIKV Asian and African lineages [23,25]. The limitation of this study was the lack of actual matrices (such as blood or CSF) to test the tolerance of the novel test on different matrices. ZIKV infections have been documented globally; however, previous epidemiological studies had not identified any cases in China until a recent ZIKV infection report emerged [34]. Acquiring clinical samples with ZIKV remains challenging; hence, this report used only viral RNA or cultured viruses for conducting limit of detection (LOD) and specificity tests on ZIKV-PAND. However, the diagnostic results using clinical simulation samples and wildtype ZIKV with ZIKV-PAND show perfect consistency with qRT-PCR assays, which can aid in molecular testing for the clinical diagnoses and epidemiological investigations of ZIKV infection.

## Figures and Tables

**Figure 1 viruses-16-00539-f001:**
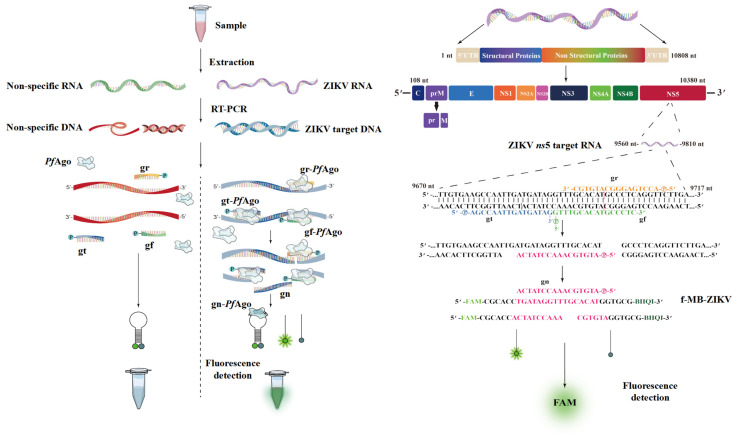
Schematic of ZIKV-PAND workflow. ZIKV-PAND, along with either the ZIKV-NS5 DNA target or its non-specific DNA control, is highlighted on the right and left sides of the left panel, respectively. After sample collection, viral RNAs were extracted and converted into target DNA with the RT-PCR step. The amplicons were targeted in the *Pf*Ago-based detection, and the results of the tests were visualized via a fluorescence assay. The nucleotide location of the NS5 target region in the ZIKV RNA genome, the size of the PCR product, the three 5’P-gDNAs (gr, gt, and gf) and the newly generated gn are indicated (**left panel**). The sequences of three 5’P-gDNAs (gr, gt, and gf), newly generated single-stranded DNA (gn), and molecular beacons are shown and highlighted in different colors (**right panel**). FAM, fluorescent dye; f-MB-ZIKV, FAM-labeled molecular beacon; gn, newly generated gDNA guide; gr, gt, and gf, three 5’P-gDNAs indicated in orange, blue, and green colors, respectively.

**Figure 2 viruses-16-00539-f002:**
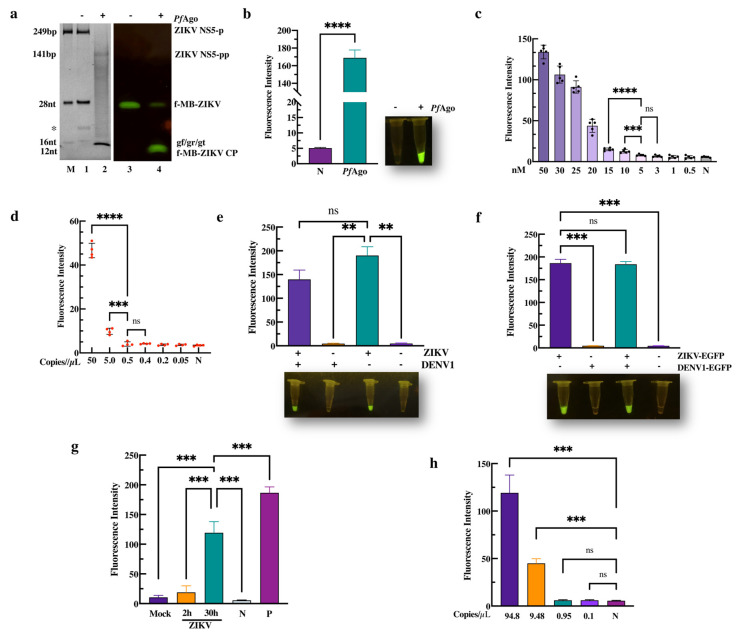
*PfAgo*-mediated Zika virus nucleotide detection. (**a**) TBE-PAGE analysis of the ZIKV-PAND system with three phosphorated gr/gf/gt stained with SYBR gold dye (Lanes 1–2), or the fluorescence image was recorded directly by UV (Lanes 3–4). The size of the target, gDNA, molecular beacon, and cleavage product bands are indicated in the left panel. The abbreviations for these components are f-MB-ZIKV CP, ZIKV NS5 PCR product cleavage product; ZIKV NS5-p, ZIKV NS5 PCR product; ZIKV NS5-pp, cleavage product of ZIKV NS5 PCR product. The marker, denoted as M, contains ZIKV NS5-p, f-MB-ZIKV, and various gDNAs (gf/gr/gt). Asterisk in left panel indicates the band of the PCR primer (20 nt). (**b**) Fluorescence-intensity detection of ZIKV-PAND based on three phosphorated gr/gf/gt is indicated. N, no *Pf*Ago added (*n* = 3). The MDC analysis of the three gDNA-mediated ZIKV-PAND without PCR (**c**) or with PCR (**d**) in a 20 µL reaction volume are indicated; relative fluorescence intensity detection in a series of diluted ZIKV NS5 target PCR products or ZIKV NS5 target as the template. (**e**) The specificity assay of ZIKV-PAND. The indicated ZIKV NS5 target region and DENV-1 conserved region are schematized in Appendix A. ZIKV and DENV1 NS5 PCR product were determined by the subsequent PAND, as described in the Section 2 (*n* = 3). (**f**) Analysis of sample simulation with ZIKV-PAND coupled with RT-PCR was conducted. RT-PCR was firstly performed with the indicated ZIKV-EGFP and DENV1-EGFP transfected RNA samples, as described in the Section 2. Next, the RT-PCR product was used for ZIKV-PAND (*n* = 4). (**g**) ZIKV RNA was measured in supernatant taken from cultured mock- and ZIKV-infected (ZIKV 2 h/30 h) Vero cells, *dd*H_2_O (negative control, N), and pUC-ZIKV plasmid (positive control, P) using ZIKV-PAND coupled with RT-PCR (*n* = 3). (**h**) The serially diluted ZIKV cDNA samples reversed transcribed from the ZIKV RNA genome were utilized for ZIKV-PAND (*n* = 3). All data are presented as mean ± SD; Student’s *t*-test; ** *p* < 0.01, *** *p* < 0.001, **** *p* < 0.0001. ns, not significant.

## Data Availability

Data are contained within the article and Appendix A.

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
