# Peer review of "PfAgo-Based Zika Virus Detection"

_viruses, 2024, doi:10.3390/v16040539_

Round 1

Reviewer 1 Report

Comments and Suggestions for Authors

In this manuscript, chen et al. described a novel nucleic acid detection system developed for detection of ZIKV infection. They nicely showed that the ZIKV-PAND is able to effectively and specifically detect ZIKV NS5 with the addition of PfAgo. However, there are some flaws in some experimental designs, and the major concern remains whether this method is limited to a single ZIKV strain or it can be widely applied.

Key concerns to be addressed are listed below:

1. Please add some background information about NS5 in the introduction. Since this detection method is based on ZIKV NS5 sequence, please also add the rationale for targeting it.

2. Result section, paragraph 1 line 3-4 do not flow well, please fix.

3. In Figure1, the illustration need further annotation both in the actual figure as well as in the legend. This will help the audience to better understand it. For example, the left panel itself has 2 panels as well, and I believe the left is an example of non-specific DNA and the right is ZIKV-NS5 DNA, these need clarification in the figure for sure. Another example is FAM need to be explained in the legend as well.

4. In Figure 2a, what is the lane M? 

5. In Figure 2b-d, the experiments are lacking a negative control sample without ZIKV NS5 target DNA. 

In 2c, please show statistical analysis between 10nM/5nM and N to support the claim that the MDC of ZIKV-PAND is 10nM. Same principle applies to Figure 2d as well.

6. In page6, last 2 lines, this statement is not scientifically accurate and solid. Please add restrictions to this sentence since it seems rush to conclude the ZIKV-PAND detection is more specific then any RT-qPCR assay.

7. In discussion, authors briefly commented on the SHERLOCK method that can be done in 2 hours, what will be the estimate time of the ZIKV-POND detection?

8. The biggest concern for this paper is that the authors only tested their method based on one ZIKV sequence. Since ZIKV have 2 distinct lineages and numerous strains, and the strains/variants sent for testing normally remain unknown for detection purpose, it is necessary to show that this detection method at least has some degree of universal application potential. The conservativity of the targeted NS5 sequence across ZIKV strains should be analyzed and discussed as well for the same reason.

Reviewer 2 Report

Comments and Suggestions for Authors

                This is a straightforward study that reports the use of PfAGo to detect ZIKV.  The experimental design is based on established approaches and the data support the conclusions that are drawn. 

My only major concern regarding the study is that the group has already published papers on the PfAgo (PAND) technology in the detection of HPV16, SARS-2, B19 and ASFV.  Thus this manuscript is a lateral extension of the published technology and thus may not have a large impact in the field.  It would be perhaps more impactful if the authors also included an application of the technology rather than just demonstrating its sensitivity and selectivity. 

Minor Points:

1.        Pg. 6:  change a16nt to a 16nt

2.       Pg. 7:  Conclusion in the heading is misspelled

Comments on the Quality of English Language

minor edits are required, some of which are listed above

Round 2

Reviewer 1 Report

Comments and Suggestions for Authors

I appreciate the authors efforts to respond to all the questions, I believe most of my concerns have been addressed and this manuscript has improved significantly since last version. With some minor revision and text editing, This paper should be considered to be published in Viruses.

1. Please try to keep the consistency of the name ZIKV-PAND throughout the paper. I noticed both ZIKV-PAND and ZIKV PAND.

2. Please specify what is RV in Figure S4 and page 6.

3. I appreciated author's answer regarding how long it takes to conduct ZIKV-PAND. Please add a breif section discussing the time(efficiency) compared to other methods in the manuscript as well.
